# Alpha2-Adrenergic Receptors as a Pharmacological Target for Spike-Wave Epilepsy

**DOI:** 10.3390/ijms24021477

**Published:** 2023-01-12

**Authors:** Evgenia Sitnikova, Elizaveta Rutskova, Kirill Smirnov

**Affiliations:** 1Institute of the Higher Nervous Activity and Neurophysiology of Russian Academy of Sciences, Butlerova Str., 5A, Moscow 117485, Russia; 2Skolkovo Institute of Science and Technology, Bolshoy Boulevard 30, Bld. 1, Moscow 121205, Russia

**Keywords:** absence epilepsy, spike-wave discharges, thalamocortical network, adrenergic receptors, alpha2 adrenergic receptors, arousal, HCN channels, locus coeruleus

## Abstract

Spike-wave discharges are the hallmark of idiopathic generalized epilepsy. They are caused by a disorder in the thalamocortical network. Commercially available anti-epileptic drugs have pronounced side effects (i.e., sedation and gastroenterological concerns), which might result from a low selectivity to molecular targets. We suggest a specific subtype of adrenergic receptors (ARs) as a promising anti-epileptic molecular target. In rats with a predisposition to absence epilepsy, alpha2 ARs agonists provoke sedation and enhance spike-wave activity during transitions from awake/sedation. A number of studies together with our own observations bring evidence that the sedative and proepileptic effects require different alpha2 ARs subtypes activation. Here we introduce a new concept on target pharmacotherapy of absence epilepsy via alpha2B ARs which are presented almost exclusively in the thalamus. We discuss HCN and calcium channels as the most relevant cellular targets of alpha2 ARs involved in spike-wave activity generation.

## 1. Introduction

Epilepsy is a neurological disorder that causes unprovoked, recurrent seizures [1,2,3] and is associated with the occurrence of abnormal electroencephalographic (EEG) activity [1,3,4,5]. One of the most common abnormal EEG patterns is a spike-wave complex. They usually occur on trains, which are called spike-wave discharges (SWDs). In this review, we will focus on typical bilateral SWDs, which are the hallmark of idiopathic generalized epilepsy, IGE [4,5,6,7,8]. Four main syndromes of IGEs are recognized, namely childhood absence epilepsy, juvenile absence epilepsy, juvenile myoclonic epilepsy, and epilepsy with generalized tonic–clonic seizures alone [8]. IGEs is considered a disorder of neuronal cells and neuronal circuits, mostly thalamo-cortical networks (see Section 2) resulting mainly from excessive neuronal firing (i.e., hyperexcitation) and by abnormally strong synchronization in widespread neuronal populations (i.e., hypersynchronization) [2,9,10].

There is a long history of pharmacotherapy of the IGE. Table 1 summarizes current knowledge about commonly used antiepileptic drugs, and their mechanisms of action and draws attention to their undesirable side effects [10,11,12,13]. These adverse effects result from a non-selective action of antiepileptic drugs on the molecular targets. Sedation is one of the most frequent effects, because many drugs activate GABAergic receptors that are widely present all over the brain. In order to increase the effectiveness of pharmacotherapy and reduce side effects, we need substances that specifically bind molecular targets responsible for generating epileptic activity.

In this review, we focus on molecular targets of relevance to noradrenaline (NA). In order to contribute to the search for new selective antiepileptic drugs, we analyze noradrenergic modulation of spike-wave activity, discuss the separation of sedative and pro-absence effects of alpha2 AR agonists and formulate testable hypotheses about the role of alpha2 AR subtypes and their interaction with various cellular partners.

Before going into details of the molecular basis of noradrenergic modulation (Section 3 and Section 4), we introduce a general concept of neuronal network mechanisms underlying spike-wave discharges.

## 2. Neural Substrate of Spike-WAVE Discharges

### 2.1. Spike-Wave Discharges as a Product of Thalamocortical Network Dysfunction

It has been well accepted that typical SWDs are produced by highly interconnected circuitry of cortical and thalamic cells [14,15,16,17,18,19]. Ascending thalamic projections terminate at ultimately all areas of the neocortex; the neocortex projects back to the thalamus and integrates intra- and interhemispheric cortico-cortical connections This highly interconnected thalamocortical neuronal network [20] is organized in a hierarchical manner. The 1st order thalamic nuclei include relay nuclei (i.e., lateral geniculate nucleus, ventro-posteromedial, and ventro-posterolateral nuclei) and send excitatory glutamatergic terminals to the primary projection areas of the neocortex (such as visual cortex, somatosensory cortex, etc.). In the primary cortical projection areas, granular neurons of layer IV are the main target of thalamic projections, and pyramidal neurons of layer V send glutamatergic afferents to the 2nd and higher-order thalamic nuclei (such as the posterior group of nuclei, pulvinar, etc.); whereas pyramidal neurons of layer VI send feedback projections to the 1st order nuclei and modulate thalamic relay neurons (this is a “modulating” input). It appears that the 1st order thalamic nuclei innervate the primary sensory, the motor, and cingulate cortices. The higher-order thalamic nuclei innervate the associative cortex. The rest of the cortical areas receive mixed afferentation from the 1st and higher-order nuclei [20,21]. This network is more properly called the cortico-thalamo-cortical network. It produces sustained rhythmic activity controlled by the “driving” and the “modulatory” inputs from the thalamic and cortical parts. It is known that genetically predetermined molecular impairments in cortical and thalamic neurons (and glia) cause epileptic spike-wave discharges [16,18,22,23,24].

Genetic, molecular, and neuronal mechanisms underlying SWDs are summarized in several excellent reviews, for instance, by Vincenzo Crunelli and co-authors in 2002 [15] and more recently in 2020 [10]. Our current knowledge about thalamo-cortical mechanisms of SWDs has been obtained in vitro and in vivo experiments performed in rabbits, cats, monkeys, and rodents, including GAERS and WAG/Rij rats. The latter two were validated genetic rat models of generalized genetic absence epilepsy [22,25,26,27,28]. The waveform of SWDs in rats is similar to that in human patients [29]. Figure 1 shows examples of high-voltage bilaterally symmetrical SWDs in a freely moving WAG/Rij rat.

As it has been mentioned above, SWDs are generated in the network comprising the thalamic part (i.e., relay thalamic nuclei, reticular thalamic nuclei) and the cortical part [10,14,15,16,17,18,19,21,30,31]. The neocortical neurons are known to play a key role in initiating and maintenance of SWDs, therefore, the SWDs-generated network is referred to as cortico-thalamo-cortical (reviewed, for example, by [10,21]). Importantly, findings gained from animals are generally confirmed in human patients, therefore, neuronal network mechanisms of SWDs generation in human patients and in animal models are considered to be the same, i.e., [10,21,25].

### 2.2. Neuronal Mechanisms of Spike-Wave Discharges: The Role of the Thalamus

The key role of the thalamus in the pathogenesis of absence epilepsy has been acknowledged by three theoretical concepts (reviewed by [15,21]). First, the centrencephalic concept of Penfield and Jasper (1958). Second, the cortico-reticular theory of Pierre Gloor (1968). Third, the theory of the “thalamic clock” by Georgy Buzsáki (1991) [32].

The centrencephalic concept refers to a hypothetical integrative system (centrencephalon) located in the brainstem and diencephalon and has diffuse projections to the neocortex and other brain structures. The main function of the centencephalon is to coordinate consciousness; and disturbances in the centrencephalon cause absence epilepsy (or petit mal epilepsy). The anatomical substrate of the centrencephalon was defined experimentally in cats: the intralaminar thalamic nuclei (nucleus centralis medialis and nucleus reuniens), in which 3 Hz electrical stimulation elicited generalized spike-wave discharges accompanied by an absence-like behavioral state. Recently Jonas Terlau et al. (2020) [33] investigated the role of the central medial nucleus in SWDs. This thalamic nucleus is known to be involved in attention and arousal processes (refs in [19]) and could be classified as the high-order thalamic nuclei. Terlau et al. pharmacologically inactivated this nucleus in anesthetized rats (GAERS, Genetic Absence Epilepsy Rats from Strasbourg, a valid model of human absence epilepsy) and found that the amplitude of the spike component in SWDs was reduced in the neocortex (motor area), but the amplitude of the wave remained the same. Considering that the central medial nucleus is considered a part of the classical ascending reticular activating system [19], the results of Terlau et al. [33] stressed the role of the central integrative brain system, i.e., the hypothetical centrencephalon.

It is well accepted that neurons of relay thalamic nuclei operate in tonic or in bursting firing modes, depending on membrane potential and membrane conduction [14,20,34,35,36]. In the waking state, the membrane potential of relay thalamic neurons is relatively high (−55 … −65 mV—weak depolarization), and neurons fire in the tonic mode conducting afferent signals from sensory nuclei to corresponding cortical projection areas [31,36,37]. In the state of drowsiness and slow-wave sleep, the afferent inflow through the thalamus (i.e., relay nuclei) to the cortex is substantially reduced in parallel to a decrease in ascending tonic influences of the brain activating systems. This system distributes “mediators of wakefulness” (such as glutamate, acetylcholine, noradrenaline, serotonin, histamine, dopamine, and orexin) [38], which are necessary to maintain tonic firing mode. Low levels of “mediators of wakefulness” and activation of “sleep centers” in ventrolateral and median preoptic areas (VLPO/MnPO) lead to a decrease of the membrane potential of relay thalamic neurons to −65 … −75 mV. Thalamic neurons enter bursting mode [34,35,36], and are capable of generating sustained rhythmic activity as long as neuronal activity in the relay thalamic nuclei and the reticular thalamic nucleus (RTN) is synchronized.

This follows from a traditional point of view suggesting that abnormal rhythmogenesis comes from the interplay between neuronal populations in the thalamic relay nucleus and the RTN [10,16,19,21,36,37]. Relay thalamic neurons are excitatory (glutamatergic) and they are reciprocally interconnected with reticular thalamic neurons (GABA-ergic). Reticular thalamic neurons inhibit relay thalamic neurons via GABAa and GABAb receptors, and this results in the de-inactivation of the T-type Ca^2+^ current [15,31,36]. This current is activated during inhibitory postsynaptic decay on repolarization of the membrane potential, giving rise to a regenerative low-threshold Ca^2+^ spike that triggers a burst of fast spikes. This spiking activity goes back to the RTN and ascends to the cortex. In such a way, the thalamo-cortical network generates synchronized burst discharges that are temporally locked to the spike component of the SWDs [15,31,36].

The cortico-reticular theory of Pierre Gloor considers generalized SWDs as the product of a ‘corticoreticular’ system that includes both subcortical (or ‘reticular’, ‘centrencephalic’) and cortical substrates (see refs in [16,17,21]). This concept was developed in a feline generalized penicillin epilepsy model. In this model, high doses of penicillin acted as a GABA-receptor antagonist, causing overexcitation of neurons. SWDs in this model were found after intramuscular injection of a high dose of penicillin and were associated with a decrease in behavioral responsiveness similar to human “absence”. In chronically penicillin-treated cats, SWDs were reduced with anti-absence drugs, such as valproate and ethosuximide. The cortico-reticular theory suggests that SWDs are developed in the hyperexcitable cortex and RTN pacemaker neurons are necessary to trigger SWDs.

The intrathalamic interplay is the underlying idea of the theory of the “thalamic clock” by Georgy Buzsáki [32]. According to this theory, the neurons of the RTN trigger SWDs, but do not act as pacemakers. The neurons in relay thalamic nuclei are bilaterally interconnected with the RTN neurons, but the number of relay thalamic neurons greatly exceeds the number of RTN neurons. Therefore, after being triggered by RTN neurons, epileptic spike-wave activity is tremendously growing as more and more thalamic neurons are getting involved. In other words, RTN engaged the thalamo-cortical system and associated structures in epileptic spike-wave activity. The fundamentally important issues of the “thalamic clock” concept are (1) RTN neurons are not pacemakers, but are triggers of paroxysmal activity; (2) spike-wave activity is generated by the thalamic neural network ‘RTN neurons’-‘relay neurons’, but not by individual pacemaker cells.

### 2.3. Neuronal Mechanisms of Spike-Wave Discharges: The Role of Neocortex

The primary role of the neocortex in the generation of SWDs was first demonstrated in 1969 by J. Bancaud and colleagues who did unique experiments in human patients by means of depth recordings and direct electrical stimulation of the frontal lobes (see refs in [16,17]). Their *“observations suggest that spike-wave discharges are secondary to a focal discharge in the frontal cortex, which is rapidly propagated over the cortex through corticocortical pathways”* cited by Meeren et al. [17]. Later on, the groups of Hans Lüders, Ernst Niedermeyer and Mark Holmes confirmed that the cortex was primarily involved in SWDs (see refs in [17,18]).

Next, a focal cortical theory of absence epilepsy was introduced by Hanneke Meeren and coauthors in 2002 [17,30]. Their study was conducted in Wistar Albino Rats from Rijswijk (WAG/Rij rats), with a well-recognized genetic rat model of absence epilepsy [28,39]. In 2002, Meeren et al. published the results of a comprehensive study of network mechanisms responsible for the onset, generalization, and high synchrony of SWDs; they found that SWDs were initiated by the primary somatosensory cortex [30]. This finding has been confirmed in numerous studies in animal models (reviewed by Antoine Depaulis and Stéphane Charpier in 2018 [22], and also in human patients (reviewed by Vincenzo Crunelli et al. in 2020 [10]. Cortical focus is characterized by increased neuronal excitability and is prone to generate epileptic discharges. A number of in vitro studies have demonstrated defects in voltage-gated ion channels and synaptic signaling that account for epileptogenic processes in cortical focus (see review [22]). Changes on the molecular level have mostly been pronounced in deep-layer pyramidal neurons. More specifically (see references in [18,22]):(1)An increased level of mRNA of voltage-gated sodium channels Nav1.1 and Nav1.6 that conduct persistent sodium current (I_NaP_) is required for generating neuronal bursts.(2)Dysfunction of non-specific cationic current I_h_ (a ‘pacemaker’ current).(3)(Amplification of action potential-triggered dendritic Ca^2+^ spikes and an increase in burst firing.(4)Impairment of GABA-mediated inhibitory mechanisms in deep cortical layers: a loss of GABAb subunits in pyramidal neurons; a decreased function of presynaptic GABAb receptors; a reduction of the fast (presumably GABAa) component of inhibitory synaptic responses.(5)Inflammatory processes mediated by pro-inflammatory molecules, such as interleukin-1 beta, and reactive astrocytosis.

In our review in 2006 [18] we defined global and local processes underlying the generation of SWDs. Global processes were neurochemical changes in GABA-ergic and the glutamatergic systems, upregulation of sodium channels; local processes were selective deficits in intracortical GABA-ergic neurotransmission, local morphometric changes of pyramidal neurons in the focal cortical zone.

## 3. Noradrenergic Regulation of Spike-Wave Activity

To date, a large amount of data has been accumulated on the noradrenergic modulation of spike-wave activity (e.g., [40,41,42,43]). Cortico-thalamo-cortical neural circuitry is known to receive dense innervations from noradrenergic neurons [44,45,46]. Nevertheless, existing anti-absence drugs do not directly affect noradrenaline-related mechanisms or molecular targets of noradrenaline (NA). Mechanisms of noradrenergic regulation of arousal level may play a significant role in the generation of SWDs. By participating in state transitions, the NA system sets favorable conditions at molecular, cellular, and network levels for the appearance of pathological spike-wave activity. We suppose that by separating a contribution of the noradrenergic system into the sleep-wake cycling and its modulation of spike-wave activity, novel, more selective pharmacological targets could be identified.

### 3.1. Noradrenergic Brain System

The noradrenergic neurons in the brain belong to the reticular formation of the medulla oblongata and the pons. The center of noradrenergic projections lies in the *locus coeruleus* (LC). The noradrenergic system is known to be involved in arousal level regulation, attention processes, behavior optimization and stress response [47]. The LC consists of only several hundreds of neurons, but has a great influence on different brain structures that could be selective and heterogeneous [48]. The effect of NA on target cells depends on a set of presynaptic and postsynaptic ARs.

There are three types of adrenergic receptors: alpha1, alpha2 and beta [49,50,51,52]. Alpha1 and beta ARs are known to mediate excitatory action, and alpha2 ARs—inhibitory action. Furthermore, these three receptor types *“have different affinities for NA, from highest to lowest: alpha2 (∼50 nM), alpha1 (∼300 nM), and beta1-3 (∼800 nM), suggesting that cells show different responses depending on the local NA concentration they are exposed to”* (cited by Wahis et al. 2021 [52]). Different LC activity modes are associated with different types of behavior and functional states [53]. During stress, the LC switches to the tonic high activity mode; the state of active wakefulness is characterized by the burst activity of LC neurons in response to significant sensory stimuli or tonic activity associated with exploration [47,54]; activity of LC neurons during non-rapid eye movement sleep (NREM) is low, and the LC is essentially silent during rapid eye movement sleep (REM) [38].

### 3.2. Adrenergic Receptors and Sedation

Alpha2 ARs agonists (such as clonidine, xylazine, medetomidine, dexmedetomidine) have pronounced sedative, anxiolytic and analgesic effects, which has led to their widespread clinical application as well as use in veterinary medicine [55,56]. A hypnotic effect of alpha2 ARs agonists resembles NREM sleep after sleep deprivation, both by the structure of sleep (suppression of REM phase) and by activating a certain population of neurons in the preoptic area of the hypothalamus [57]. Studies in the late 60-th–70-th of the last century demonstrated that alpha2 ARs controlled the NA release from presynaptic terminals [58,59], and this explained the sedative effect of alpha2 AR agonists. For instance, clonidine microinjection into the LC led to a decrease in LC activity [60] and sedation [61]. This effect seemed to be associated with the direct inhibition of the LC neurons through dendritic and somatic alpha2 ARs, and not with a change in NA concentration through autoreceptors in the axonal terminals. Indeed, intravenous administration of even small doses of clonidine led to an immediate decrease in the activity of the LC neurons [62], while a drop in NA levels in the brain could be postponed [63].

Recently, evidence has been accumulating that the activation of postsynaptic alpha2 ARs played a major role in sedation and analgesia. Selective knockout of presynaptic receptors in noradrenergic neurons as well as chemical destruction of noradrenergic cells did not lead to a loss of the sedative effect of clonidine or medetomidine [64,65,66]. Moreover, mice lacking NA due to knockout of gene coding dopamine β-hydroxylase showed hypersensitivity to dexmedetomidine-induced sedation. This unexpected result could be possibly explained by taking into account the fact that in control mice the presence of NA-activated excitatory alpha1 and beta receptors, which could counteract the effect of alpha2 AR agonists. Thus, mice lacking NA demonstrated only the pure effect of alpha2 activation and thus a faster transition to sleep, which was described in the paper by Hu et al. [67].

### 3.3. Adrenergic Receptors and Spike-Wave Activity

The influence of the noradrenergic system on spike-wave activity has been shown in several rat models of absence epilepsy (summarized in Table 2). In general, it was found that beta ARs were not implicated in the genesis of SWDs. Activation of alpha2 ARs enhances spike-wave seizures as well as the inhibition of alpha1 ARs. The reviews of Giorgi et al. [68] and Fitzgerald et al. [69] summarized the pro- and antiepileptic effects of NA and drugs that affected certain types of noradrenergic receptors, as well as the activity of the LC, the center of the NA system.

The modulation of spike-wave activity through alpha2 ARs seems to be the most pronounced. Noteworthy, the prototypical alpha2 AR agonist clonidine has a pro-epileptic effect in human patients and in genetically predisposed rats [42,67,68]. Considering the fact that clonidine increases spike frequency in patients with drug-resistant epilepsy, it seems to be a useful tool to induce pathological EEG activity for diagnostic purposes [70,71].

**Table 2 ijms-24-01477-t002:** Noradrenergic modulation of spike-wave discharges and similar phenomena.

Model	Type of Treatment	Target	Effect	Reference
Tottering mice	6-OHDA (s.c., dose 100 mg/kg) Acute effect on the first or second day after birth	Noradrenergic terminals (hyperinnervation) + Noradrenergic terminals (destruction)	Decrease in total duration of SWDs	Noebels, 1984 [72]
GAERS rats	Salbutamol (i.p., 1.25–50, acute) Isoprenaline (i.p., 12.5–100, acute)	Beta AR (activation)	No effect	Micheletti et al., 1987 [40]
	Propranolol (i.p., 1.25–80, acute)	Beta AR (inhibition)	No effect	
	Prazosin (i.p., 0.25–4, acute)	Alpha1 AR (inhibition)	Increase in SWDs total duration	
	ST 587 (i.p., 1–4, acute) Cirazoline (i.p., 0.1–4, acute)	Alpha1 AR (activation)	Decrease in SWDs total duration	
	Clonidine (i.p., 0.01–0.1, acute)	Alpha2 AR (activation)	Increase in SWDs total duration	
	Yohimbine (i.p., 0.5–8, acute)	Alpha2 AR (inhibition)	<2 mg/kg: decrease in SWD total duration >4 mg/kg: very short decrease in SWDs total duration followed by the disappearance of the effect	
	Desipramine (i.p., 5–40, acute)	NA reuptake (inhibition) 5HT reuptake (inhibition)	No effect	
	Mianserine (i.p., 1.25–40, acute)	NA release (activation) alpha1 AR (inhibition) alpha2 AR (inhibition) 5HT receptors (inhibition)	No effect	
Fischer 344 rats	6-OHDA (i.c., 200 μg, two injections with 48 h interval)	Noradrenergic terminals (destruction)	Increase in HVS incidence (4–7 days after administration)	Buzsáki et al., 1991 [41]
	6-OHDA (i.th., 50 μg, one injection)		Increase in HVS incidence (4–7 days after administration)	
	DSP-4 (i.p., 50 mg/kg, acute)		No effect (4–7 days after administration)	
	+ Xylazine (i.th., acute)		Increase in HVS incidence after intracisternal or intrathalamic 6-OHDA	
	Clonidine (i.p., 0.02 or 0.1 mg/kg)	Alpha2 AR (activation)	Increase in HVS incidence	
	Xylazine (i.p., 0.5 or 2 mg/kg)	Alpha2 AR (activation)	Increase in HVS incidence	
	Yohimbine (i.p., 1 or 5 mg/kg, acute)	Alpha2 AR (inhibition)	Decrease in HVS incidence 1 mg/kg: maximal effect	
	Prazosin (i.p., 0.5 or 2 mg/kg)	Alpha1 AR (inhibition)	Increase in HVS incidence	
	Desipramine (i.p., 1 or 10 mg/kg, acute) Amitriptyline (i.p., 1 or 10 mg/kg, acute)	NA reuptake (inhibition) 5HT reuptake (inhibition)	Decrease in HVS incidence	
	Amitriptyline (i.p., 10 mg/kg, 21 days)	NA reuptake (inhibition) 5HT reuptake (inhibition) alpha2 AR (decreased density)	Decrease in HVS incidence Decrease of the effect of intrathalamic xylazine Decrease of the effect of IP xylazine—not significant	
	Clonidine (i.th., bilateral, 0.1 or 1 nmol, acute)	Alpha2 AR (activation)	Increase in HVS incidence (suppressed by 5 nmol of yohimbine)	
	Clonidine (i.th., unilateral, 10 or 100 pmol, acute)	Alpha2 AR (activation)	Increase in HVS amplitude	
WAG/Rij rats	Clonidine (i.p., 0.00625 mg/kg, acute)	Alpha2 AR (activation)	Increase in SWDs incidence Decrease in total EEG power in the frontal cortex Increase in total EEG power in RTN Decrease in intracortical coherence	Sitnikova and Luijtelaar 2005 [42]
	Dexmedetomidine (i.p., 1 mg/kg, IP, acute)	Alpha2 AR (activation)	Decrease in total SWDs number (very high dose)	Al-Gailani et al., 2022 [73]
GAERS rats	Atipamezole (i.c.v., 1–31 µg, acute)	Alpha2 AR (inhibition)	12/31 µg: decrease in SWDs incidence and SWDs mean duration	Yavuz et al., 2020 [43]
	Atipamezole (i.c.v., 12 µg, 5 days)	Alpha2 AR (inhibition)	Decrease in total SWDs duration	
	Dexmedetomidine (i.c.v., 0.1, 0.5, 2.5 µg, acute)	Alpha2 AR(activation)	Increased in total SWD, absence status epilepticus	Yavuz et al., 2022 [74]
Charles River rats	Clonidine (p.o., 0.0001–0.1 mg/kg, acute)	Alpha2 AR (activation)	Increase in the mean duration of SWDs	Kleinlogel, 1985 [75]
	Guanfacine (p.o., 0.0001–0.1 mg/kg, acute)	Alpha2 AR (activation)	Increase in the mean duration of SWDs	
	Yohimbine (p.o., 0.1–10 mg/kg, acute)	Alpha2 AR (inhibition)	Decrease in the mean duration of SWDs (maximal effect with dose 1 mg/kg). 3.2 mg/kg: suppressed the effect of guanfacine (1 mg/kg)	
	Prazosin (p.o., 0.32–10 mg/kg, acute)	Alpha1 AR (inhibition)	Increase in the mean duration of SWD	
Long-Evans rats	Yohimbine (i.p., 0.5–10 mg/kg, acute)	Alpha2 AR (inhibition)	0.5–5 mg/kg: decrease in the mean duration of FEAD 10 mg/kg: no effect	King and Burnham, 1982 [76]
Wistar rats	Atipamezole (s.c., 0.1/1/10 mg/kg, acute)	Alpha2 AR (inhibition)	0.1 mg/kg: no effect 1/10 mg/kg: suppression of HVS	Riekkinen et al., 1990 [77]
	Guanfacine (i.p., 0.004/0.02/0.1 mg/kg, acute)	Alpha2 AR (activation)	Increase in HVS incidence and duration (0.004 mg/kg: no effect on duration) + Atipamezole (1 mg/kg): suppressed an increase in HVS duration Atipamezole (10 mg/kg): suppressed an increase in HVS duration and incidence	
	+ unilateral RT lesion (VB also damaged)		No HVS on the contralateral side; HVS still occurred on the ipsilateral side	
	Atipamezole (s.c., minipump, 0.1 mg/kg/h, continuous)	Alpha2 AR (inhibition)	Decrease in HVS incidence during the 6-day administration No changes in sensitivity to Guanfacine (i.p., 0.001 mg/kg, acute)	Jäkälä et al., 1992 [78]
Aged Wistar rats (10–12 months)	Atipamezole (i.p., 0.01–4 mg/kg, acute)	Alpha2 AR (inhibition)	Decrease in HVS incidence	Yavich et al., 1994 [79]
	Idazoxan (i.p., 0.1–4 mg/kg, acute)	Alpha2 AR (inhibition)	<0.5 mg/kg: decrease in HVS incidence >0.5 mg/g: disappearance of the effect	
	Yohimbine (i.p., 0.1–4 mg/kg, acute)	Alpha2 AR (inhibition)	<0.5 mg/kg: decrease in HVS incidence >0.5 mg/g: disappearance of the effect	
	Dexmedetomidine (i.p., 0.005 mg/kg, acute)	Alpha2 AR(activation)	Increase in HVS incidence	
	Prazosin (i.p., 1 mg/kg, acute)	Alpha1 AR (inhibition)	Increase in HVS incidence	

Abbreviations: AR—adrenoreceptor, NA—noradrelaine, 5HT—serotonin. Type of EEG activity: FEAD—flash-evoked after discharge, HVS—high voltage spindles (the waveform similar to spike-wave discharges), SWDs—spike-wave discharges. Type of administration: i.p.,—intraperitoneal, i.c.—intracisternal; i.th.—intrathalamic; i.c.v.—intracerebroventricular, PO—perioral, SC—subcutaneous.

### 3.4. Separation of Proepileptic and Sedative Effects of Alpha2 ARs Activation

George Buzsáki et al. suggested that spike-wave activity can be regulated by postsynaptic alpha2 ARs in the thalamus [41]. They showed that intrathalamic injection of the alpha2 ARs agonist xylazine increased the incidence of neocortical high-voltage spindles even after the destruction of noradrenergic terminals [41]. At the same time, local injection of clonidine into the thalamus at a dose, which resulted in sedation when administered into LC, was inefficient to sedate rats [58]. This fact together with our own observations brought us to the idea about the separability of sedative and pro-absence pathways activated by alpha2 ARs agonists.

SWDs much more often occur during behavioral inactivity and drowsiness (i.e., [77]), while our observations indicated that immediately after dexmedetomidine injection and later, following the phase of deep sedation, SWDs interrupted active behavior. Appendix A shows two episodes of SWDs in a freely moving rat immediately after i.p. dexmedetomidine injection (dose 0.05 mg/kg) and after drug elimination (3 h after injection). Such a behavioral pattern is not typical for animal models of absence epilepsy [25,28]. Recently, Melis Yavuz et al. [74] found that intracerebroventricular injection of the agonist of alpha2 AR, dextometomidine, in GAERS rats resulted in continuous SWDs resembling absence status epilepticus. These authors defined two sets of absence statuses: the first—1–2 min after dextometomidine injections, and the second—after dexmedetomidine-induced sleep. They concluded that sleep and absence status epilepticus in dexmedetomidine-injected rats were completely separate, therefore, it was concluded that this pharmacological model cam be “*a tool to investigate the sleep and absence status transitions*” [74].

In naturally falling asleep, the NA level gradually decreases [80] and the predominantly alpha1- and beta-AR-mediated excitatory effect of NA switches to an alpha2-mediated inhibitory effect. This state is favorable for triggering spike-wave activity [28,77] (the upper part in Figure 2) Natural awakening provides an almost instant turn to arousal with a very low probability of SWDs occurring [28,81]. Injections of alpha2 ARs agonists might cause an unnaturally rapid increase in activation of alpha2 ARs leading to an artificial predominance of alpha2-mediated effect over the effect of other ARs subtypes (the bottom part of Figure 2). The emergence from sedation is again an extended transition favorable for SWDs to occur. Future studies of pharmacologically induced absence status epilepticus are required to examine the dose-dependent effects of alpha2 ARs agonists on SWDs. Another intriguing issue regarding the role of each ARs subtypes in the modulation of spike-wave activity and arousal level.

Figure 3 demonstrates expression profiles of alpha2 AR subtypes in the areas of the rat brain involved in the generation of SWDs. There are three subtypes of alpha 2 ARs: A, B, C. Studies that analyzed mRNA, as well as protein content, revealed that the largest number of alpha2A ARs is located in the LC, where they serve as autoreceptors limiting NA release and controlling neuronal firing. Although to a lesser extent, they are found in many other parts of the brain, in particular in the cerebral cortex with a gradient increasing to the deep layers. Subtype A is also expressed in most of the nuclei of sleep- and wake-promoting systems (such as ventrolateral preoptic area and median preoptic area, *substantia nigra* and dorsal and lateral hypothalamus, parabrachial nuclei, dorsal raphe nuclei) [82] highlighting its role in sedative and anesthetic effects [83,84,85]. Indeed, it has been shown that sedative and analgesic effects of nonselective alpha2 AR agonists are mostly provided by subtype A [86,87].

The functions of subtypes B and C are less understood so far. The highest density of subtype C is detected in the striatum where it modulates the release of GABA [88] and in the cerebral cortex [83,85,89]. Subtype C is known to inhibit NA release, but to a lesser extent than subtype A [90]. Mice with overexpressed alpha2C ARs showed an impairment in searching behavior during water maze training [91]. More recently some possibilities of therapeutic interventions through alpha2C ARs were discovered since its selective antagonist showed antidepressant and antipsychotic effects [92,93]. It is known that subtypes A and B undergo pronounced desensitization after 30 min of exposure to NA, while subtype C becomes less sensitive to its endogenous agonist only after 24 h [94]. Receptor internalization in the presence of agonists also differs between subtypes, with subtype C being the least prone to it [95]. Therefore, the impact of alpha2C ARs may become more pronounced after a prolonged increase in endogenous NA or after non-selective alpha2 AR agonist exposure.

Currently, one of the few established physiological effects of subtype B knockout in the central nervous system is the absence of nitric oxide-induced antinociception [96]. In the light of noradrenergic regulation of spike-wave activity, important is the fact that the subtype Alpha2B is found almost exclusively in the thalamus, which receives dense NA innervations [97]. Exceptional is the reticular nucleus, where alpha2B ARs are not present [83]. Alpha2B ARs are the main postsynaptic receptors in the thalamus, since the presence of its mRNA has been confirmed by various studies, while there are contradictory results about the other AR subtypes [85,98,99]. The predominance of alpha2 AR in the thalamus over the other brain subtypes suggests that activation of thalamic alpha2B ARs results in an increase of SWDs in animals with destructed noradrenergic terminals after local administration of alpha2 ARs agonists [41]. Altogether suggests the possibility for selective alpha2B AR antagonists as selectively acting drugs for the treatment of absence epilepsy.

Non-selective alpha2 ARs antagonists suppress spike-wave activity [40,43], however, their low specificity can lead to undesirable effects, such as enhanced sexual behavior, decreased locomotor activity, altered tactile sensitivity, impaired cognitive functions, and increased blood pressure [100,101,102]. A study of acute behavioral effects of alpha2B AR antagonists is in demand to reveal if their side effects will not neutralize the possible suppression of absence epilepsy.

## 4. Cellular Targets of Alpha2 ARs in Relation to Spike-Wave Activity

The influence of alpha2 ARs is mainly carried out through Gi/o-proteins although coupling to Gs was also demonstrated [103]. Thus, activation of alpha2 ARs can either inhibit or stimulate various intracellular pathways. Numerous neuronal proteins interacting directly or indirectly with alpha2 ARs have been described. Here we focus on phenomena related to spike-wave activity, such as interaction with ionic channels.

### 4.1. Alpha2-Adrenoreceptors and HCN Channels

HCN channels control neuron excitability and participate in the stabilization of resting potential [104]. Due to their unique ability to open during hyperpolarization creating an inward current named I_h_, activation of HCN channels may depolarize the neuron enough to initiate rebound bursting, a form of spiking activity crucial for spike-wave discharges [105]. There are four HCN isoforms (HCN1–4). HCN1 isoform is mainly expressed in the cerebral cortex, and HCN2 is mostly in the thalamus but also in the cortex, and HCN4 in the thalamus (with the exception of the reticular nucleus) but not in the cortex. HCN3 is expressed at a low level in both structures and thus will not be analyzed further [105,106,107,108]. HCN2 channels activate slower and at more negative voltages and have an increased sensitivity to cAMP modulation than HCN1 [109,110]. An increase in cAMP leads to a more pronounced shift in the activation of HCN2 channels into the depolarized direction, which leads to the elimination of differences in the degree of hyperpolarization needed to activate both isoforms. HCN4 has the slowest kinetics and is strongly modulated by cAMP [111,112]. Melis Yavuz and Filiz Onat in 2018 published a comprehensive review about the role of HCN channels in the pathophysiology of absence epilepsy (also in rat models), where they emphasized the involvement of the second messenger system in modulating HCN channels [113].

In different models of absence epilepsy and under different registration conditions, sometimes opposite changes in the HCN channels were found [114,115,116]. However, it can be concluded that blocking the I_h_ current in the cortex leads to an increase in the excitability of neurons [117]. This effect is based on enhanced temporal summation of the distal dendritic excitatory postsynaptic potentials, which can participate in SWDs generation [118,119]. Conversely, blocking the I_h_ current in the thalamus suppresses burst firing and SWDs [120]. HCN channels have an increasing distal expression gradient on dendrites (that is, the farther away from the soma, the denser) [121], thus they provide signal filtering by weakening the contribution of remote excitatory postsynaptic potentials, EPSPs [122].

The interaction of alpha2 ARs and HCN channels was studied in the context of epileptogensis [122,123] and pain relief [124]. It has been shown that activation of postsynaptic alpha2 ARs blocks I_h_ current, which increases input resistance and enhances temporal summation during trains of distally evoked EPSPs making a cell more excitable [125]. Whether this mechanism is sufficient to aggravate spike-wave activity is yet to be investigated. Studies of the combined effects of the local application of substances acting on alpha2 ARs and HCN channels may clarify the role of their interaction in the spike-wave initiating site in the somatosensory cortex.

If alpha2 AR agonists inhibit HCN, then injection of alpha2 AR agonists into the thalamus should have an anti-absence effect, because blockage of the I_h_ current in thalamic neurons suppresses SWDs [109]. This contradicts the results of Buzsáki et al. (1991) [41]. Therefore, the effect of alpha2 AR agonists on I_h_ current may differ in the thalamus and in the cortex [117]. Differences in the molecular cascades triggered by the activation of alpha2 AR subtypes are mostly studied in cell expression systems and need to be proved in vivo. Nevertheless, activation of the alpha2B subtype, predominantly expressed in the thalamus, may have a stimulatory effect on cAMP synthesis in contrast to the inhibitory action of alpha2A ARs [126]. Selective coupling of alpha 2-adrenergic receptor subtypes to cyclic AMP-dependent reporter gene expression in transiently transfected JEG-3 cells.

There is contradictory data on whether alpha2 AR agonists affect HCN channels through the modulation of cAMP synthesis or other pathways such as activation of protein kinase C [43] or even by direct binding in an alpha2 AR-independent manner (Figure 4) [43,127,128,129]. The resulting effect seems to be dependent on which subtype of alpha2 and which isoforms of HCN interact. For example, increasing cAMP counteracts the inhibitory effect of ZD7288, a potent blocker of I_h_, or dexmedetomidine and guanfacine on HCN channels [128,130,131]. Thus, it may increase or even reverse inhibitory effects in tissues where cAMP-sensitive isoforms of HCN are highly expressed. Another way of alpha2-mediated upregulation of HCN channels is cellular alkalinization [132] which shifts I_h_ activation potential to more depolarized values [133]. A recent study showed that in rats genetically predisposed to absence epilepsy, spike-wave activity can be triggered by hypoxia-induced blood alkalosis resulting in the activation of neurons in the intralaminar thalamic nuclei [134].

It should be mentioned that the impact of I_h_ on neuron excitability is intertwined with currents mediated by other ion channels such as G protein-coupled inwardly rectifying K+ channels [135], which themselves may be influenced by alpha2 AR agonists. Dexmedetomidine activates a G protein-coupled inwardly rectifying K+ current [136] which is suppressed by the first choice anti-absence drug ethosuximide [137]. Considering the impending revision of the mechanism of action of ethosuximide as an anti-absence drug [137], it is interesting to compare its effects on cell currents with the effects of alpha2 AR antagonists.

### 4.2. Alpha2-Adrenoreceptors and Calcium Channels

Calcium channels are important regulators of neuronal firing properties (reviewed in [138]). Calcium channels are divided into low-voltage activated (T-type) and high-voltage activated (L-, N-, P/Q-, R-types). The role of calcium channels in the absence epilepsy is confirmed by genetic findings in patients and studies in knockout animal models. Pharmacological modulation of the spike-wave activity using calcium channel blockers and activators has been shown in animals genetically predisposed to absence seizures. In particular, intraperitoneal and intracerebroventricular microinjections of L-type blockers resulted in an increase in the number and duration of SWDs, but intracortical microinjections suppressed SWDs [139,140].

Systemic administration of L-current activator BAY K8644 decreased the number and duration of spike-wave discharges, regardless of the method of administration. L-type calcium channel blockers are known to modulate the sedative effects of alpha-2 AR agonists. Local administration of nifedipine into LC and a subhypnotic dose of dexmedetomidine led to the loss of righting reflex [141]; intraperitoneal injection of a low dose of nifedipine (2 mg/kg) blocked the sedative effect of clonidine [142], and the high dose (20 mg/kg) enhanced dexmedetomidine-induced sleep time [143]. Nevertheless, the sedative effects of alpha2 AR agonists were at least partially mediated by the blocking of L-type calcium channels, since the injection of nifedipine restored the hypnotic ability of dexmedetomidine in rats who developed tolerance after chronic administration [143]. Noteworthy, the method of administration determined how an L-current blocker would affect a decrease in blood pressure induced by clonidine. Intravenous injection of nifedipine induced an increase in blood pressure in rabbits, but intravenous administration prevented the hypotensive action of clonidine [144]. Since L-type blockers are used as antihypertensive drugs, their action on spike-wave activity and locomotion after systemic administration might be accounted for a decrease in blood pressure. Indeed, SWDs in humans and animals are known to be linked with a reduction in blood pressure [145]. Identification of the contribution of L-current into pro-absence and sedative effects of alpha2 AR activation is possible with the local co-administration of substances into the central nervous system.

Alpha2 agonists reduce N- and P/Q-type currents [103,146,147]. There is only limited evidence of the effects of N- or P/Q blockers on spike-wave activity, but they were either indistinguishable from the general detrimental influence of N-blockers or minor, as in the case of P/Q blockers [139]. However, since mutations in high-voltage gated calcium channels lead to the development of absence epilepsy phenotypes in animal models [148,149], the study of alpha2 ARs activation or inhibition in these models may indicate other ways to modulate spike-wave activity.

Altogether, T-type calcium channels play a crucial role in the generation of SWDs [150,151], and interactions between alpha2 ARs and T-type calcium channels need to be further investigated. An increase in thalamic burst firing mediated by T-type channels might not be as significant in the generation of SWDs as previously assumed [152,153]. Therefore, future studies are required to examine the relationship between the burst firing of thalamic cells and the SWD-promoting effects of alpha2 AR agonists.

### 4.3. Astrocytic Alpha2-Adrenoreceptors

More and more evidence has been obtained in favor of the fact that astrocytes can play an important role in the pathogenesis of epilepsy [154,155]. Most studies have demonstrated the involvement of astrocytes in the pathogenesis of focal epilepsies [154,155,156,157], but recently the role of astrocytes in the pathogenesis of absence epilepsy has been disclosed. Optogenetic excitation of astrocytes in the ventrobasal thalamus caused a pro-absence effect in GAERS and WAG/Rij rats [158].

Astrocytes might be a promising target for the therapy of absence epilepsy. Alpha2A ARs are highly expressed in astrocytes, where they potentiate glutamate synthesis [159] and stimulate GABA release [160]. Although the general increase in GABA mediates the therapeutic effects of valproate, the administration of GABAa or GABAb agonists into the ventrobasal thalamus leads to an increase in spike-wave activity [161,162]. Given the special role of astrocytes in controlling the level of GABA in the thalamus, where they provide the main contribution to the reuptake of this neurotransmitter [163], they can be suggested as a target for novel drugs. It can be assumed that blocking thalamic alpha2 ARs with the selective antagonist to subtype B can reduce the level of GABA and prevent hyperpolarization of relay neurons, which creates a predisposition for burst firing.

## 5. Conclusions

Spike-wave discharge occurrence is a result of disrupted interactions between the cortex and the thalamus. Activation of alpha2 ARs by both systemic and intrathalamic administration leads to increased spike-wave activity, and this effect is predominantly mediated by postsynaptic receptors. In animal models, spike-wave activity occurs in a state of inactive wakefulness, often preceding sleep. Immediately after the administration of alpha2 AR agonists, rats enter a behavioral state similar to the state of ‘absence’ in patients with IGE (accompanied by spike-wave discharges) that is interrupted by episodes of active behavior. This indicates that the neural network involved in spike-wave activity generation in animal models, although being connected to the system regulating sleep and wakefulness, can be separated from it. Nonselective antagonists of alpha2 ARs suppress spike-wave activity in animal models. However, their use can lead to various undesirable behavioral and cognitive consequences. Understanding how various effects mediated by alpha2 receptors are carried out is important for the development of more selective drugs. A possible way to provide the selectivity of the effects is to activate or block certain subtypes of alpha2 ARs, considering their different expression profiles in brain structures involved in sleep initiation and spike-wave generation. The almost exclusive expression of subtype B in the thalamus indicates that it is this particular subtype activation that provides the pro-absence effect of intrathalamic administration of non-selective alpha2 AR agonists. We suggest that spike-wave activity could be decreased by introducing selective alpha2B ARs antagonists, while brain functions executed by other structures would not be affected.

Activation of alpha2 ARs influences several ion currents, whose changes have been reported in animal models of absence epilepsy. Blocking of HCN channels in cortical neurons by alpha2 ARs agonists administration leads to an increase in their input resistance and widening of a window of temporal EPSPs summation. That in turn makes the excitability of cortical neurons higher and could contribute to a pro-absence effect. In addition, aggravation of spike-wave discharges can be carried out by impacting calcium channels and regulating GABA levels by astrocytes. A better understanding of noradrenergic modulation of spike-wave activity can contribute to the development of new selective drugs with higher benefits and fewer side effects.

## Figures and Tables

**Figure 1 ijms-24-01477-f001:**
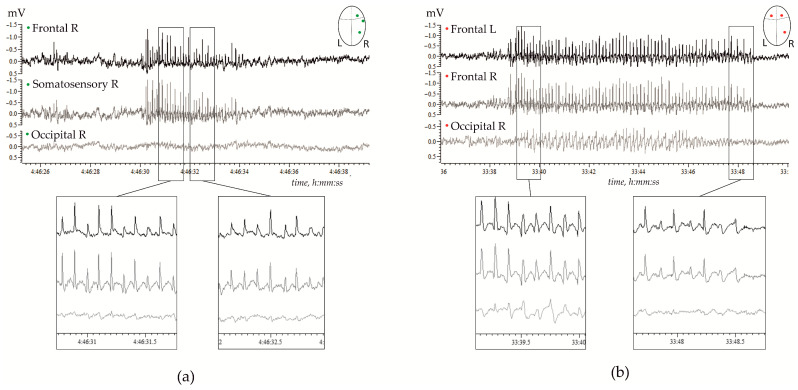
Examples of spontaneous spike-wave discharges (SWDs) in electrocorticograms recorded in freely moving adult WAG/Rij rats; (**a**)—11 months old subject, and (**b**)—7 months old subject. (**a**) Regular high-voltage 8–10 Hz SWDs were expressed in the right hemisphere (R) over the frontal and parietal (somatosensory) cortical areas, but were hardly seen over the occipital area. (**b**) High-voltage 8–10 Hz SWDs were bilaterally synchronized, and were present in the left (L) and right frontal cortical areas, and were seen in the occipital cortex.

**Figure 2 ijms-24-01477-f002:**
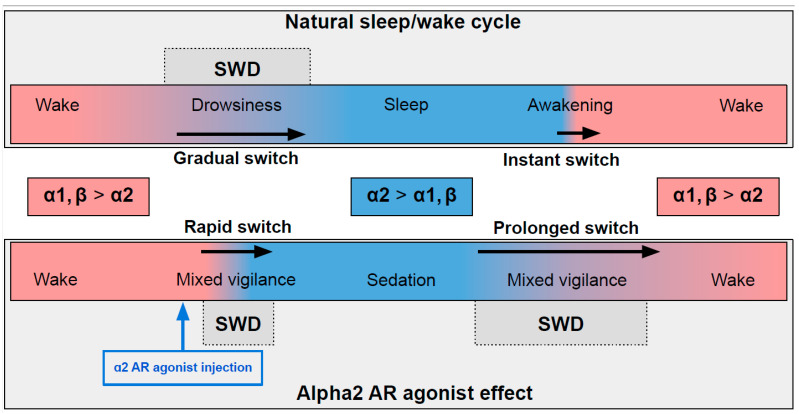
The schema demonstrating adrenergic mechanisms of sleep modulation and spike-wave discharge (SWD) modulation. The upper plot demonstrates the natural drug-free state. The bottom plot—pharmacologically induced condition after administration of agonist of alpha2 adrenoreceptors. Noradrenaline affects alpha1, alpha2 and beta-receptors, and NA concentration during sleep/sedation is lower (blue area) than during wakefulness (rose area). Transient states between wake and sleep/sedation are favorable for SWD to occur. See explanations in the text.

**Figure 3 ijms-24-01477-f003:**
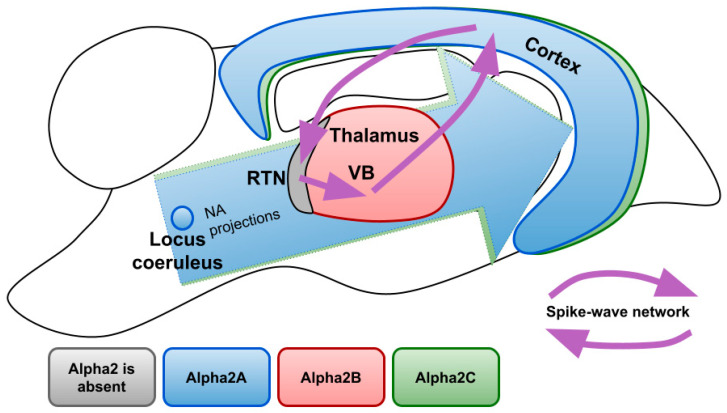
Expression profiles of different alpha2 AR subtypes over the brain structures involved in the generation of SWDs. NA—noradrenaline; RTN—reticular nucleus; VB—ventrobasal thalamus.

**Figure 4 ijms-24-01477-f004:**
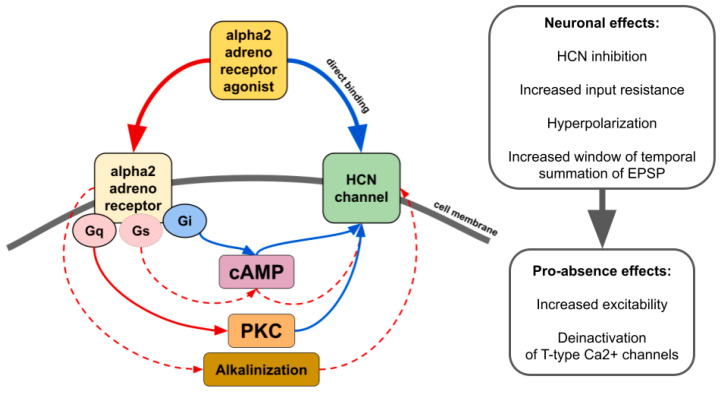
The main effects of alpha2 AR agonists on HCN channels. Red lines—activation. Blue lines—inhibition. Solid lines—effects have strong empirical evidence. Dashed lines—effects have limited evidence.

**Table 1 ijms-24-01477-t001:** Antiepileptic drugs commonly used for treatment of idiopathic generalized epilepsy (IGE). The summary from books and reviews Crunelli et al. 2020 [10], Shorvon 2011 [11], Gören and Onat 2007 [12] and, Brigo et al. 2021 [13].

Drug	Usage	Action	Side Effects
Ethosuximide (ETX)	The drug of choice in IGE	Reduces low threshold T-type Ca^2+^ currents in thalamic neurons	Dose-dependent side effects are related to the gastrointestinal tract (i.e., hematopoietic adverse effects) or the central nervous system (a wide variety of idiosyncratic reactions). Nausea, abdominal discomfort, vomiting, diarrhea, and anorexia.
		Decreases the persistent Na^+^ and Ca^2+^-activated K^+^ currents in thalamic and cortical neurons	
		Reduces cortical GABA levels in cortical neurons.	
		Reduces elevated glutamate levels in cortical neurons	
Valproic acid (VPA)	The drug of first choice for many types of epilepsy, including IGE	Act on GABAa receptors. 1. Increases GABA concentrations in synaptosomes via activation of the GABA—synthesizing enzyme [glutamic acid decarboxylase]. 2. Inhibits GABA catabolism through inhibition of GABA transaminase and succinic semialdehyde dehydrogenase.	Nausea, vomiting, hyperammonaemia and other metabolic effects, endocrine effects, severe hepatic toxicity, pancreatitis, drowsiness, cognitive disturbance, aggressiveness, tremor, weakness, encephalopathy, thrombocytopenia, neutropenia, aplastic anemia, hair thinning and hair loss, weight gain, polycystic ovarian syndrome. Teratogenicity.
		Inhibits excitatory neurotransmission mediated by aspartic acid, glutamic acid and γ—hydroxybutyric acid.	
		Reduces conductance at the voltage—dependent Na^+^ channels. Reduction of the threshold for Ca^2+^ and K^+^ conductance in the hippocampus.	
Lamotrigine	Effective in generalized tonic–clonic, typical and atypical absence seizures. Add-on or monotherapy of focal seizures and generalized seizures.A second-line drug, reserved for intractable absence seizures.	Blockage of voltage—dependent Na^+^ channel conductance (similar to carbamazepine or phenytoin). Suggested actions: 1. Anti-glutamate and anti-aspartate effects. 2. Modulation of the glycine-binding site on the NMDA receptor. 3. Modulation of voltage-dependent Ca-conductance at N-type Ca-channels and K^+^ conductance.	Rash (sometimes severe), blood dyscrasia, headache, ataxia, asthenia, diplopia, nausea, vomiting, dizziness, somnolence, insomnia, depression, behavioral effects, psychosis, tremor. Marked risk of hypersensitivity.
Carbamazepine (CBZ)	The use in IGE is more controversial. It may exacerbate myoclonus, generalized absence seizures and other non—convulsive types.	Blockage of neuronal Na^+^ channels, pre- and post-synaptically. Blockade of the Na^+^ channels is believed to inhibit glutamate release. Inhibitor of NMDA receptors, Agonists of gamma-aminobutyric acid (GABA) agonist.	Sedation, fatigue, diplopia, headache, depression, dizziness, nausea, and ataxia. Acute hypersensitivity (skin), a dose-related antidiuretic effect
Levetiracetam	A wide range of seizure types (with the partial onset and generalized) and with good efficacy in myoclonus and absence seizures. It is effective as monotherapy and adjunctive therapy.	The antiepileptic action is not fully understood. It binds selectively and with high affinity to SV2A (a synaptic vesicle protein that is involved in synaptic vesicle exocytosis and presynaptic neurotransmitter release). It has a neuroprotective potential.	Somnolence, asthenia, infection, dizziness, headache, irritability, aggression, behavioral and mood changes, emotional lability, depersonalization, psychosis, nervousness, seizure exacerbation, rhinitis, cough, vomiting
Phenobarbital (PB)	Monotherapy and adjunctive therapy of generalized seizures (including absence and myoclonus) in adults and children.	GABAa-receptor agonist. Reduction of glutamate excitability, affecting K^+^, Na^+^ and Ca^2+^ conductance	Sedation, ataxia, dizziness, insomnia, hyperkinesis (children), dysarthria, mood changes (especially depression), behavior change, aggressiveness, cognitive dysfunction, impotence, reduced libido, folate deficiency and megaloblastic anemia, vitamin K and vitamin D deficiency, osteomalacia, Dupuytren contracture, frozen shoulder, shoulder—hand syndrome, connective tissue abnormalities, rash. Risk of dependency. Potential for abuse
Primidone (PRM)	A prodrug of phenobarbital with probably some minor additional efficacy. Monotherapy and adjunctive therapy in generalized tonic–clonic seizures	Same to Phenobarbital	Intense dizziness, nausea and sedation. Fewer behavioral side effects than either phenobarbital or phenytoin
Benzodiazepines (BZDs):	Diazepam (DZP) Lorazepam (LZP) Midazolam (MZL) Acute treatment of absence status epilepticus	A positive allosteric modulator of GABAa receptors. Enhancement of inhibitory neurotransmission.	Sedation, addiction, development of tolerance

Abbreviations: IGE—idiopathic generalized epilepsy; GABA—γ-aminobutyric acid (inhibitory mediator); NMDA receptor-N-methyl-d-aspartate (glutamate receptor).

## Data Availability

Not applicable.

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
