# Peer review of "Alpha2-Adrenergic Receptors as a Pharmacological Target for Spike-Wave Epilepsy"

_ijms, 2023, doi:10.3390/ijms24021477_

Round 1

Author Response

Dear Reviewer,

Thank you for your positive review. We highly appreciate the time and effort that you dedicated to evaluating our paper. Your comments were about the clarity of some statements and about Figure 2. Please find our point-to-point answers below.

Line 17, could mention not only adrenergic receptor subtypes but also pre or post synaptic localization/ cell type

Mentioning activation of particular receptors in the abstract, we mean receptors of particular brain networks. In very deed, every process in the brain is characterized by a unique combination of parameters on the network, cellular and molecular levels. We discuss mechanisms of spike-wave discharges and sedation in depth in the other parts of our review.

Line 73, can also mention cortico-cortico connections, particularly innervation by interhemispheric ones.

In the revised version, we stressed the importance of cortico-cortical connections in Lines 66-68: “the neocortex projects back to the thalamus and integrates intra- and interhemispheric cortico-cortical connections, thus composing a This highly interconnected thalamocortical functional neuronal network [20].”

Figure 2 is a bit confusing and may require a different structure.

Figure 2 has been updated and refined with improvements in structure.

Reviewer 2 Report

The authors have a comprehensive review of the adrenergic mechanisms and neuronal and cortical theories behind the spike-and-wave discharges of the absence epilepsy. Alpha-adrenergic regulation seems to be one of the main mechanisms behind the SWD generation and maintenance, therefore this review is highlighting this issue. Here are some remarks that may improve the manuscript.

·      The sentence in the abstract is not clear. ‘’alpha2 ARs agonists provoke sedation and enhanced spike-wave activity during transitions to and from a sedation deep phase’’. 

·      On table 1, the references need to be cited for each mechanism of action. The targets and actions were proved in various studies, that might have some discrepancies in the future, so it will be important to know the actual source of the study.

·      In table 1, while the target of ethosuximide is given in detail with anatomic locations and specific neuronal projections, the target of other antiepileptics are not given this slightly disturbs the flow of information. 

·      In table 1 other mechanisms of actions of the other antiepileptics are not given. For example carbamazepine increases GABA and decreases glutamate.

·      Which combination therapy the authors mean for ethosuximide usage?

·      Authors state that the ‘’ARs are inhibitory and the most sensitive to NA. ‘’ Any relevant reference?

·      On page 9 this discussion point has some contradictions: ‘’Moreover, mice lacking NA due to knockout of gene coding dopamine β-hydroxylase showed hypersensitivity to dexmedetomidine-induced sedation. This unexpected result could be possibly explained taking into account the fact that the basal presence of NA activated excitatory alpha1 and beta receptors, which could counteract the effect of alpha2 AR agonists. Thus, mice lacking NA demonstrated only the pure effect of alpha2 activation and thus a faster transition to sleep, which was described in the paper by Hu et al. [64].’’ For instance, theoretically with a knockout model, there should not be noradrenaline without the synthesizing enzyme, how the authors explain any basal levels of noradrenaline? Also the reason of the singular activation of alpha2 receptors should not be due to the lack of noradrenaline levels.

·      On page 11, there is a spelling error at the dose of 12 µg of atipamezole.

·      Recent preprinted paper states dexmedetomidine-induced absence status epilepticus/continuous SWD activity in another genetic absence epilepsy model, GAERS, this information can also be given in the table 1 and the following paper should be cited. preprinted article: Melis Yavuz, SERDAR AKKOL, Filiz Onat. ALPHA-2A ADRENERGIC RECEPTOR (α2AR) ACTIVATION IN GENETIC ABSENCE EPILEPSY: AN ABSENCE STATUS MODEL?. Authorea. December 07, 2022. DOI: 10.22541/au.167042890.02021046/v1

·      On page 13 this discussion point is not clear: ‘’George Buzsáki et al. suggested that spike-wave activity can be regulated by 291 postsynaptic alpha2 ARs in the thalamus [41]. They showed that intrathalamic injection 292 of the alpha2 ARs agonist xylazine increased the incidence of neocortical high-voltage 293 spindles even after the destruction of noradrenergic terminals [41]. At the same time, local 294 injection of clonidine into the thalamus at a dose, which resulted in sedation when 295 administered into LC, was inefficient to sedate rats [58]. This fact together with our own 296 observations brought us to the idea about the separability of sedative and pro-absence pathways activated by alpha2 ARs agonists’’. Neocortical high voltage spindle are not fuly grown spike-and-wave discharges. Therefore it may still point out a overlap of sedation and wave complex of SWDs.

·      On page 13 authors state that ‘’Such a behavioral pattern, which is not typical for animal models of absence epilepsy, can be caused by a rapid increase in activation of alpha2 ARs accompanied by the residual activation of alpha1 and beta ARs due to a non-instantaneous decrease in NA level.’’ Is there a relevant reference explaining the occupancy of all these receptors might trigger sleep/wake switch or the authors believe this might be the case? Any proof of concept for this mechanism?

·      From the video, the switch between the sedation and SWD, as well as movement and SWD sometimes can be seen also in animals recovering from anesthesia, different types. There is also a limited time-resolution of the switch between states, meaning the baseline EEG activity of wakefulness may occur in between sedation and SWD. The authors might want to comment on this limitation. 

·      Also the above-mentioned study shows an immediate time-resolution of icv delivered dexmedetomidine, the instatenous and with a high time-resolution of the switch, the authors might also want to discuss this difference in between two delivery routes (Melis Yavuz, SERDAR AKKOL, Filiz Onat. ALPHA-2A ADRENERGIC RECEPTOR (α2AR) ACTIVATION IN GENETIC ABSENCE EPILEPSY: AN ABSENCE STATUS MODEL?. Authorea. December 07, 2022. DOI: 10.22541/au.167042890.02021046/v1). In Figure 4, the transition of sleep-and-status/continuous SWD is clearly seen in an exact switch manner, unlike the IP data with dexmedetomidine in WAG/Rij, that the authors have submitted-figure 2). Mentioning the switch mechanism might necessitate citing this paper again.

·      The authors state that the thalamus only has the alpha2b subtype, but there are studies indicating that the thalamus might have other isoforms as well. Some of them lower but present (On page 15 ‘’ Since the only ARs subtype present in thalamus is alpha2B, …’’). Here are some references for that:

a.     Talley EM, Rosin DL, Lee A, Guyenet PG, Lynch KR. Distribution of alpha 2A-adrenergic receptor-like immunoreactivity in the rat central nervous system. J Comp Neurol. 1996 Aug 12;372(1):111-34. doi: 10.1002/(SICI)1096-9861(19960812)372:1<111::AID-CNE8>3.0.CO;2-6. PMID: 8841924.

b.     Tavares A, Handy DE, Bogdanova NN, Rosene DL, Gavras H. Localization of alpha 2A- and alpha 2B-adrenergic receptor subtypes in brain. Hypertension. 1996 Mar;27(3 Pt 1):449-55. doi: 10.1161/01.hyp.27.3.449. PMID: 8698452.

c.     Arponen E, Helin S, Marjamäki P, Grönroos T, Holm P, Löyttyniemi E, Någren K, Scheinin M, Haaparanta-Solin M, Sallinen J, Solin O. A PET Tracer for Brain α2C Adrenoceptors, (11)C-ORM-13070: Radiosynthesis and Preclinical Evaluation in Rats and Knockout Mice. J Nucl Med. 2014 Jul;55(7):1171-7. doi: 10.2967/jnumed.113.135574. Epub 2014 May 5. PMID: 24799619.

·      The modulation of HCN channels through alpha2a AR has been discussed in a genetic absence epilepsy model in a review, hypothesising HCN channel modulation through alpha2a can be inhibitory in SWDs DOI: 10.14744/epilepsi.2018.03371. In the study the figure also illustrates the interaction of alpha2-HCN channels.

·      The idea of specific blockage of cortical HCN leading to increase in SWDs ‘’However, it can 387 be concluded that blocking the Ih current in the cortex leads to an increase in the 388 excitability of neurons. ‘’ as well as ‘’Therefore, the effect of alpha2 AR agonists on Ih current may differ in the thalamus and in 407 the cortex. ‘’ were previously discussed in the following paper. Yavuz M, Aydın B, Çarçak N, Onat F. Decreased Hyperpolarization-Activated Cyclic Nucleotide-Gated Channel 2 Activity in a Rat Model of Absence Epilepsy and the Effect of ZD7288, an Ih Inhibitor, on the Spike-and-Wave Discharges. Pharmacology. 2022;107(3-4):227-234. doi: 10.1159/000520059. Epub 2022 Jan 10. PMID: 35008085.

Overall the idea behind the alpha2b subtype should be drug targets only depends the localized expressions in thalamus. But considering the publications that may show otherwise, raise questions of the strength of the idea of this specific subtype. The authors might consider the redirection of the idea instead covering all subtypes for now with the current knowledge.

Minor issues:

The tables and the space between the columns can be readjusted in the tables, as well as justifying the sentences will be better looking

Spelling error -- GABAa instead of GABAA,  GABAb as well.

Author Response

Dear Reviewer,

Thank you for your positive feedback and for highlighting the strong points of our study. We highly appreciate your efforts and time in evaluating our paper. Thank you for your suggestions on ways to improve clarity, and for other relevant references. We revised our review in accordance with your critical remarks.

The sentence in the abstract is not clear. ‘’alpha2 ARs agonists provoke sedation and enhanced spike-wave activity during transitions to and from a sedation deep phase’’. 

This sentence has been revised: “alpha2 ARs agonists provoke sedation and enhance spike-wave activity during transitions from awake/sedation”

On table 1, the references need to be cited for each mechanism of action. The targets and actions were proved in various studies, that might have some discrepancies in the future, so it will be important to know the actual source of the study.

Table 1 has just a supplementary role in demonstrating that anti-absence medication has considerable side-effects. This Table has been shortened. Almost each mechanism requires all references, e.g. [10-13]. Therefore we put them in Table title. “The summary from books and reviews Crunelli et al 2020 [10], Shorvon 2011 [11], Gören & Onat 2007 [12] and Brigo et al 2021 [13]”.

In table 1, while the target of ethosuximide is given in detail with anatomic locations and specific neuronal projections, the target of other antiepileptics are not given this slightly disturbs the flow of information. 

Table 1 has been shortened, and one column (targets) has been removed.

In table 1 other mechanisms of actions of the other antiepileptics are not given. For example carbamazepine increases GABA and decreases glutamate.

Thank you for this comment. We have added information about mechanisms of action of carbamazepine. Another anti-epileptic drug, phenytoin, has not been included in Table 1. As to Lamotrigine and Levetiracetam, mechanisms of their anti-epileptic effect have not been fully understood.

Which combination therapy the authors mean for ethosuximide usage?

We simplified the sentence: “The drug of choice in IGE” (deleted “combination therapy” and “monotherapy”).

Authors state that the ‘’ARs are inhibitory and the most sensitive to NA. ‘’ Any relevant reference?

We totally revised this paragraph. Lines 247-252: “There are three types of adrenergic receptors: alpha1, alpha2 and beta [49-52]. Alpha1 and beta ARs are known to mediate excitatory action, and alpha2 ARs - inhibitory action. Furthermore, these three receptor types “have different affinities for NA, from highest to lowest: alpha2 (50 nM), alpha1 (300 nM), and beta1-3 (800 nM), suggesting that cells show different responses depending on the local NA concentration they are exposed to” (cited by Wahis et al 2021 [52]).”

On page 9 this discussion point has some contradictions: ‘’Moreover, mice lacking NA due to knockout of gene coding dopamine β-hydroxylase showed hypersensitivity to dexmedetomidine-induced sedation. This unexpected result could be possibly explained taking into account the fact that the basal presence of NA activated excitatory alpha1 and beta receptors, which could counteract the effect of alpha2 AR agonists. Thus, mice lacking NA demonstrated only the pure effect of alpha2 activation and thus a faster transition to sleep, which was described in the paper by Hu et al. [64].’’ For instance, theoretically with a knockout model, there should not be noradrenaline without the synthesizing enzyme, how the authors explain any basal levels of noradrenaline? Also the reason of the singular activation of alpha2 receptors should not be due to the lack of noradrenaline levels.

There was an error in the description of results: the basal level of NA was measured in control mice. This sentence has been corrected (Lines 282-283): “This unexpected result could be possibly explained taking into account the fact that in control mice the basal presence of NA activated excitatory alpha1 and beta receptors.”

On page 11, there is a spelling error at the dose of 12 µg of atipamezole.

The paper of Yavuz et al, 2020 said that the minimum SWD-suppressive dose of atipamezole was 12 µg.

Recent preprinted paper states dexmedetomidine-induced absence status epilepticus/continuous SWD activity in another genetic absence epilepsy model, GAERS, this information can also be given in the table 1 and the following paper should be cited. preprinted article: Melis Yavuz, SERDAR AKKOL, Filiz Onat. ALPHA-2A ADRENERGIC RECEPTOR (α2AR) ACTIVATION IN GENETIC ABSENCE EPILEPSY: AN ABSENCE STATUS MODEL?. Authorea. December 07, 2022. DOI: 10.22541/au.167042890.02021046/v1

Thank you for the reference to this relevant study. We cited this paper in Table 2 and also in Lines 326-333.

On page 13 this discussion point is not clear: ‘’George Buzsáki et al. suggested that spike-wave activity can be regulated by 291 postsynaptic alpha2 ARs in the thalamus [41]. They showed that intrathalamic injection 292 of the alpha2 ARs agonist xylazine increased the incidence of neocortical high-voltage 293 spindles even after the destruction of noradrenergic terminals [41]. At the same time, local 294 injection of clonidine into the thalamus at a dose, which resulted in sedation when 295 administered into LC, was inefficient to sedate rats [58]. This fact together with our own 296 observations brought us to the idea about the separability of sedative and pro-absence pathways activated by alpha2 ARs agonists’’. Neocortical high voltage spindle are not fuly grown spike-and-wave discharges. Therefore it may still point out a overlap of sedation and wave complex of SWDs.

There is a crucial difference between high voltage spindles (HVS) and sleep spindles. George Buzsáki et al. studied mechanisms of HVS [e.g., Buzsáki, G.; Kennedy, B.; Solt, V.B.; Ziegler, M. Noradrenergic Control of Thalamic Oscillation: The Role of Alpha-2 Receptors. Eur. J. Neurosci. 1991, 3, 222–229, doi:10.1111/j.1460-9568.1991.tb00083.x. and Buzsaki, G., Bickford, R. G., Ponomareff, G., Thal, L. J., Mandel, R. and Gage, F. H. (1988b) Nucleus basalis and thalamic control of neocortical activity in the freely moving rat. J. Neurosci., 8, 4007-4026]. The frequency of HVS (around 8 Hz) was lower than in sleep spindles (12-14 Hz). HVS contained very high and repetitive spikes, in contrast to smooth sleep spindles. HVS were suppressed by ethosuximide, therefore HVS in Buzsáki et al. studies could be considered as analogs of SWDs.

On page 13 authors state that ‘’Such a behavioral pattern, which is not typical for animal models of absence epilepsy, can be caused by a rapid increase in activation of alpha2 ARs accompanied by the residual activation of alpha1 and beta ARs due to a non-instantaneous decrease in NA level’’ Is there a relevant reference explaining the occupancy of all these receptors might trigger sleep/wake switch or the authors believe this might be the case? Any proof of concept for this mechanism?

We are sorry for the lack of clarity in our concept. We reformulated it in Lines 325-355. "In naturally falling asleep, NA level gradually decreases [80] and the predominantly alpha1- & beta-AR-mediated excitatory effect of NA switches to alpha2-mediated inhibitory effect. This state is favorable for triggering spike-wave activity [28, 77] (the upper part in Fig. 2) Natural awakening provides an almost instant turn to arousal with a very low probability of SWDs to occur [28, 81]."

From the video, the switch between the sedation and SWD, as well as movement and SWD sometimes can be seen also in animals recovering from anesthesia, different types. There is also a limited time-resolution of the switch between states, meaning the baseline EEG activity of wakefulness may occur in between sedation and SWD. The authors might want to comment on this limitation. 

The video has been changed. Now it shows SWDs occurring at baseline and after i.p. administration of Dextometomidine (in dose 0.05 mg/kg). Two minutes after Dextometomidine injections, SWDs occurred during wakefulness and were associated with a behavioral arrest. The same phenomenon was noted 3-6 hours after this Dextometomidine injection. We have not analyzed switches between waking state to SWDs and back.

Also the above-mentioned study shows an immediate time-resolution of icv delivered dexmedetomidine, the instatenous and with a high time-resolution of the switch, the authors might also want to discuss this difference in between two delivery routes (Melis Yavuz, SERDAR AKKOL, Filiz Onat. ALPHA-2A ADRENERGIC RECEPTOR (α2AR) ACTIVATION IN GENETIC ABSENCE EPILEPSY: AN ABSENCE STATUS MODEL?. Authorea. December 07, 2022. DOI: 10.22541/au.167042890.02021046/v1). In Figure 4, the transition of sleep-and-status/continuous SWD is clearly seen in an exact switch manner, unlike the IP data with dexmedetomidine in WAG/Rij, that the authors have submitted-figure 2). Mentioning the switch mechanism might necessitate citing this paper again.

Thank you for this idea. The rout of drug administration is indeed important, and Melis Yavuz et al (2022) described the almost immediate effect of i.c.v. injected Dextometomidine on SWDs in GAERS. We observed the same rapid effect after systemic injections of Dextometomidine. It seems also important that spike-wave seizures were more intensive after emergence from sedation. This has also been mentioned by Yavuz et al (2022). In Lines 369-377 we added: “Recently, Melis Yavuz et al [74] found that intracerebroventricular injection of the agonist of alpha2 AR, dextometomidine, in GAERS rats resulted in continuous SWDs resembling absence status epilepticus. These authors defined two sets of absence status: the first - 1-2 minutes after dextometomidine injections, and the second – after dexmedetomidine-induced sleep. They concluded that sleep and absence status epilepticus in dexmedetomidine-injected rats were completely separate, therefore, it was concluded that this pharmacological model cam be “a tool to investigate the sleep and absence status transitions” [74].

The authors state that the thalamus only has the alpha2b subtype, but there are studies indicating that the thalamus might have other isoforms as well. Some of them lower but present (On page 15 ‘’ Since the only ARs subtype present in thalamus is alpha2B, …’’). Here are some references for that:

  1. Talley EM, Rosin DL, Lee A, Guyenet PG, Lynch KR. Distribution of alpha 2A-adrenergic receptor-like immunoreactivity in the rat central nervous system. J Comp Neurol. 1996 Aug 12;372(1):111-34. doi: 10.1002/(SICI)1096-9861(19960812)372:1<111::AID-CNE8>3.0.CO;2-6. PMID: 8841924.
  2. Tavares A, Handy DE, Bogdanova NN, Rosene DL, Gavras H. Localization of alpha 2A- and alpha 2B-adrenergic receptor subtypes in brain. Hypertension. 1996 Mar;27(3 Pt 1):449-55. doi: 10.1161/01.hyp.27.3.449. PMID: 8698452.
  3. Arponen E, Helin S, Marjamäki P, Grönroos T, Holm P, Löyttyniemi E, Någren K, Scheinin M, Haaparanta-Solin M, Sallinen J, Solin O. A PET Tracer for Brain α2C Adrenoceptors, (11)C-ORM-13070: Radiosynthesis and Preclinical Evaluation in Rats and Knockout Mice. J Nucl Med. 2014 Jul;55(7):1171-7. doi: 10.2967/jnumed.113.135574. Epub 2014 May 5. PMID: 24799619.

Thank you for pointing out this shortcoming. We read the literature and updated citations. Lines 395-402: “Alpha2B ARs are the main postsynaptic receptors in the thalamus, since the presence of its mRNA has been confirmed by various studies, while there are contradictory results about the other AR subtypes [85,98,99]. The predominance of alpha2 AR in the thalamus over the other brain subtypes suggests that activation of thalamic alpha2B ARs results in an increase of SWDs in animals with destructed noradrenergic terminals after local administration of alpha2 ARs agonists [41]. Altogether suggests the possibility for selective alpha2B AR antagonists as selectively acting drugs for the treatment of absence epilepsy.”

Modulation of HCN channels through alpha2a AR has been discussed in a genetic absence epilepsy model in a review, hypothesising HCN channel modulation through alpha2a can be inhibitory in SWDs DOI: 10.14744/epilepsi.2018.03371. In the study the figure also illustrates the interaction of alpha2-HCN channels.

Thank you for this reference. We mentioned this study in the revised manuscript. Lines 424-427 “Melis Yavuz and Filiz Onat in 2018 published a comprehensive review about the role of HCN channels in the pathophysiology of absence epilepsy (also in rat models), where they emphasized involvement of the second messenger system in modulating of HCN channels [113].”

The idea of specific blockage of cortical HCN leading to increase in SWDs ‘’However, it can 387 be concluded that blocking the Ih current in the cortex leads to an increase in the 388 excitability of neurons. ‘’ as well as ‘’Therefore, the effect of alpha2 AR agonists on Ih current may differ in the thalamus and in 407 the cortex. ‘’ were previously discussed in the following paper. Yavuz M, Aydın B, Çarçak N, Onat F. Decreased Hyperpolarization-Activated Cyclic Nucleotide-Gated Channel 2 Activity in a Rat Model of Absence Epilepsy and the Effect of ZD7288, an Ih Inhibitor, on the Spike-and-Wave Discharges. Pharmacology. 2022;107(3-4):227-234. doi: 10.1159/000520059. Epub 2022 Jan 10. PMID: 35008085.

We are sorry for overlooking this study. This reference has now been added.

Overall the idea behind the alpha2b subtype should be drug targets only depends the localized expressions in thalamus. But considering the publications that may show otherwise, raise questions of the strength of the idea of this specific subtype. The authors might consider the redirection of the idea instead covering all subtypes for now with the current knowledge.

We agree that this statement is too straightforward. There is contradictory data on the presence of alpha2A and alpha2C mRNA and proteins in the thalamus, while the alpha2B mRNA and protein are consistently observed. We have removed categoricity from our formulation, however, we believe that this subtype may play a major role in spike-wave aggravation.

Minor issues:

The tables and the space between the columns can be readjusted in the tables, as well as justifying the sentences will be better looking

Tabulation in tables has now been refined.

Spelling error -- GABAa instead of GABAA,  GABAb as well.

It is corrected.